# The Prejudice Towards People with Mental Illness Scale: Psychometric Properties of the Italian Version (PPMI-IT)

**DOI:** 10.3390/ejihpe15070126

**Published:** 2025-07-07

**Authors:** Francesca Bruno, Francesco Chirico, Hicham Khabbache, Younes Rami, Driss Ait Ali, Valentina Cardella, Maria Chayinska, Ivan Formica, Amelia Rizzo

**Affiliations:** 1Department of Cognitive Sciences, Pedagogical Psychological and Cultural Studies, University of Messina, 98122 Messina, Italy; francesca.bruno3@studenti.unime.it (F.B.); valentina.cardella@unime.it (V.C.); maria.chayinska@unime.it (M.C.); ivan.formica@unime.it (I.F.); 2Post-Graduate School in Occupational Health, Catholic University of the Sacred Heart, 00168 Rome, Italy; medlavchirico@gmail.com; 3Faculty of Literature and Human Sciences, Sidi Mohamed Ben Abdellah University, Fez 30000, Morocco; hichamcogn@gmail.com (H.K.); younes.rami@usmba.ac.ma (Y.R.); driss.aitali@usmba.ac.ma (D.A.A.); 4Medical-Legal Center of Messina, National Institute of Social Welfare, 98122 Messina, Italy

**Keywords:** PPMI, prejudice, mental illness, mental health

## Abstract

Currently, there are no validated instruments in Italian specifically designed to assess mental illness stigma or prejudice. Moreover, implicit measures, while insightful, are often resource-intensive and impractical for large-scale population studies of Italian speakers. The present study investigated the validity of the Italian version of the Prejudice towards People with Mental Illness scale (PPMI-IT) in measuring biases toward individuals with mental health issues. The original instrument by Kenny et al. was translated from English into Italian and vice versa. A sample of 455 Italian-speaking participants (65% female; M_age_ = 33.39; SD = 13.21) was utilized to conduct a confirmatory factor analysis, confirming a four-factor structure (*fear/avoidance, malevolence, authoritarianism, unpredictability*). Factor loadings indicated that each dimension was well represented, supporting the construct validity of the scale. Model fit indices, including chi-square (χ^2^ = 782.54, df = 296.00, χ^2^/df = 2.64), RMSEA (0.06, 90% CI: 0.060–0.07), CFI (0.93), TLI (0.91), and SRMR (0.06), suggest an excellent model fit. Furthermore, the analysis of correlations and the heterotrait/monotrait (HTMT) ratio provides evidence supporting the discriminant validity of the PPMI scale compared with social desirability. These findings confirm that the PPMI scale is a valid and reliable tool for assessing biases toward individuals with mental health issues, making it suitable for academic research, clinical interventions, and public policy contexts.

## 1. Introduction

### 1.1. Criticisms in the Measurement of Prejudice

Prejudice is a complex social phenomenon that significantly impacts individuals and communities, influencing attitudes, behaviors, and societal structures. Studying prejudice is essential for addressing issues of discrimination and inequality, promoting inclusion, and improving interpersonal and group dynamics. However, examining prejudice poses several challenges due to its multifaceted nature and the factors influencing its expression and measurement.

Prejudice often operates on both explicit and implicit levels. While explicit prejudice refers to consciously held biases, implicit prejudice involves unconscious attitudes and stereotypes. These dual levels require different approaches to measurement and interpretation.

As a consequence, studies addressing prejudice toward individuals with mental health conditions suffer from theoretical, conceptual, and psychometric challenges.

To bridge these gaps, [14] ([14]) introduced the Prejudice towards People with Mental Illness (PPMI) scale, offering a refined framework that integrates stigma and prejudice research. Stigma includes elements like stereotypes and discrimination, yet prejudice, often defined as a negative attitude, remains underexplored and poorly measured in this context.

Numerous scales have been developed to assess stigma toward individuals with mental illness, many of which include items designed to measure prejudice. Researchers have highlighted the importance of construct validity, which ensures that a scale accurately measures the intended psychological construct and aligns with its theoretical antecedents and consequences ([9]). Despite these concerns, many existing scales fail to clearly define the constructs they measure and often combine elements like beliefs, attitudes, and stereotypes about mental illness ([29]; [12]).

For instance, some scales do not distinguish between awareness of stereotypes and approval of them, and their items tend to focus on multiple aspects of mental illness, such as the person, the illness, or the treatment, without addressing the evaluative component central to prejudice. This lack of clarity impedes the connection between these measures and discriminatory behavior. Moreover, many scales suffer from methodological issues, such as double-barreled items (questions that combine two ideas), and often fail to account for response biases, such as acquiescence, which can distort results ([11]).

Furthermore, many scales do not adequately address social desirability bias, which can influence how individuals report their attitudes. This is particularly problematic when measuring negative attitudes toward people with mental illness, as individuals may be hesitant to express prejudiced views due to social expectations. To mitigate this, researchers frequently construct balanced scales with an equal number of positively and negatively keyed items.

Individuals often modify their responses to align with socially acceptable norms, a phenomenon known as social desirability bias. This is particularly problematic when studying prejudice, as respondents may under-report biases or discriminatory behaviors due to fear of judgment or repercussions ([3]).

The PPMI scale is particularly well suited to address critical gaps in existing instruments for measuring mental illness stigma due to its conceptual specificity, multidimensional structure, and empirical rigor. Unlike more generic stigma measures, which often conflate mental illness with other forms of social deviance or rely on outdated dichotomies (e.g., dangerous vs. non-dangerous), the PPMI was explicitly designed to capture the unique contours of prejudice directed toward individuals with mental disorders. Its four theoretically grounded dimensions—fear and avoidance, malevolence, authoritarianism, and unpredictability—reflect a refined and differentiated understanding of how stigma operates across emotional, cognitive, and behavioral domains. This structure allows researchers to disentangle different facets of prejudice, offering a more precise diagnostic tool for both theoretical inquiry and intervention design. Moreover, the scale’s psychometric validation ensures reliability across diverse populations, making it a robust alternative to instruments that lack cultural or contextual sensitivity. In this sense, the PPMI does not merely replicate existing tools but advances the field by offering a more comprehensive and theoretically coherent framework for the empirical study of mental illness stigma.

### 1.2. The National Context

The research on stigma related to mental illness in Italy highlights a complex panorama where qualitative methods, such as narrative reviews and structured tools like questionnaires and interviews, emerge as key approaches to understanding and addressing the issue ([23]; [8]).

One example is the Italian adaptation of the Attribution Questionnaire-27 (AQ-27-I), developed by [19] ([19]), which uses a vignette-based format to assess public stigma across multiple dimensions, including responsibility, pity, anger, fear, and avoidance. Respondents are presented with a *hypothetical scenario*—typically involving a character named *Harry* who has schizophrenia—and are asked to rate their emotional and behavioral responses.

Narrative reviews broadly explore perceptions and attitudes toward mental illness, synthesizing available evidence ([21]). These studies, often based on data from clinical and community contexts, underline how stigma is rooted in cultural, social, and professional factors ([17]; [25]). The qualitative approach has demonstrated that educational interventions targeting mental health awareness among young people reveal the effectiveness of initiatives aimed at reducing prejudice and promoting positive attitudes ([18]; [2]). However, these studies are not based on the psychometric approach and lack specific instruments and measures, despite being conducted in Italy.

Studies employing qualitative structured tools, such as tailored questionnaires and interviews, provide a more detailed understanding of personal and professional experiences. Some studies based on interview techniques used with healthcare workers and caregivers highlight the weight of perceived and internalized stigma, both in hospital and family settings. Ad hoc questionnaires, used to measure implicit and explicit attitudes, demonstrate how stigma affects clinical decisions and the quality of relationships between patients and medical staff ([5]; [13]), but without providing validity tests.

A central theme in the national literature is also the comparison between the perception of stigma associated with traditional mental illnesses, such as bipolar or psychotic disorders, and more recent conditions, like substance use disorders. Studies conducted in Italy use mixed methodologies to analyze these aspects, revealing that terminology and labeling models play a crucial role in perpetuating or mitigating prejudice ([4]; [1]). In this case, the fact that the researchers had to independently construct measures highlights once again that a tool in the Italian language is not available.

### 1.3. Present Study Aim

An Italian version of the PPMI scale is, hence, necessary because no comparable instrument exists in the Italian context to measure prejudice toward individuals with mental health issues. The present study fills a critical gap by providing a validated and culturally adapted tool that addresses this need.

We aim to conduct an extensive validation process, including descriptive statistics, reliability analysis, confirmatory factor analysis (CFA), construct validity, and discriminant validity assessments.

These comprehensive analyses would ensure that the Italian version of the PPMI scale is psychometrically robust and culturally relevant, providing researchers and practitioners with a valuable tool to assess and address mental health stigma in the Italian context.

## 2. Materials and Methods

### 2.1. Sample

To validate a scale comprising 28 items in the general population, it is recommended to recruit a sample size following the rule of thumb of 5–10 participants per item ([30]). This approach ensures sufficient statistical power for exploratory and confirmatory factor analyses. Accordingly, a minimum sample size of 140 participants is required, while an ideal sample size would be 280 participants. A total of 455 individuals participated in the study, with excellent sampling adequacy (KMO = 0.927): 131 males, 316 females, and 8 nonbinary, aged between 18 and 75 years (65% female; M_age_ = 33.39; SD = 13.21. Of the sample, 40.2% reported being employed, 32.3% were students, and 18.2% were both studying and working.

### 2.2. Procedure

The tests were selected and organized in accordance with conventionally accepted ethical standards, and the participants provided informed consent and were given preliminary information about the study. Participation was voluntary and anonymous. The tests were administered via Google Forms during the period from February to June 2023. The link was promoted through social media platforms such as Instagram, Facebook, Twitch, Telegram groups, and WhatsApp.

Participants were also asked to provide demographic information such as age, gender (including perceived gender), and life activity (work, study, etc.), as well as whether they had ever received a psychiatric diagnosis or knew someone with a mental disorder. This information was collected to identify potential differences in prejudice measurement.

### 2.3. Ethical Approval

The research followed the Ethical Guidelines of the Helsinki Declaration and the Ethical Guidelines for Internet Research (NESH), and was approved by the Ethical Committee of the Polish Society of Disaster Medicine (protocol n. 16.01.2023.IRB).

### 2.4. Materials

#### The Prejudice Towards People with Mental Illness (PPMI) Scale

The PPMI scale consists of 28 items and uses a 9-point scale ranging from 0 (strongly disagree) to 8 (strongly agree). [14] ([14]) designed and validated the scale by identifying four dimensions (Table 1) that underlie prejudice toward people with mental illnesses: (1) *fear and avoidance*, which assesses fear and the tendency to avoid establishing an approach or relationship with people suffering from mental disorders; (2) *malevolence*, which assesses feelings of contempt and antipathy toward people with mental disorders; (3) *authoritarianism*, which assesses the tendency to restrict the freedom of people with mental disorders; (4) *unpredictability*, which assesses beliefs about the predictability or unpredictability of behaviors exhibited by people with mental disorders.

### 2.5. Translation Back into English

Following the guidelines from [24] ([24]), an external individual, proficient in English, was contacted to assist with the translation of the scale. The original version of the scale was not provided to them. A comparison revealed that out of a total of 352 words, only 15 differed (representing just 4.2% of the total), while 95.74% of the words were identical. The entire procedure is detailed in the Appendix A, Appendix B and Appendix C. The scale showed a readability score (Flesch Reading Ease) of 60, suggesting its applicability to the general population.

#### 2.5.1. Balanced Inventory of Desirable Responding—Italian Version (BIDR-6)

The BIDR-6 questionnaire was translated and adapted to the Italian context through the back translation method by two independent researchers with the support of a native English speaker. A 6-point Likert scale was prepared, ranging from 1 (strongly disagree) to 6 (strongly agree), without a neutral midpoint to force respondents to express judgments about themselves.

The BIDR-6 consists of 16 items, aimed at investigating 2 dimensions: (1) self-deceptive enhancement, i.e., the dynamics of self-deception; (2) impression management, i.e., the process through which individuals attempt to manipulate the impression they leave on others, related to social desirability ([10]).

#### 2.5.2. Reliability of the Scale

[14] ([14]) reported the following Cronbach’s alpha scores for the four subscales: fear/avoidance (α = 0.89), malevolence (α = 0.73), authoritarianism (α = 0.72), and unpredictability (α = 0.86). The subdimensions were moderately to strongly intercorrelated, with the strongest correlation between fear/avoidance and authoritarianism (r = 0.64), and the weakest between malevolence and unpredictability (r = 0.31).

In the present study, the alpha values for the same dimensions of PPMI-IT are: fear/avoidance (α = 0.89), malevolence (α = 0.80), authoritarianism (α = 0.85), and unpredictability (α = 0.85). Thus, greater reliability is observed in this study for the subscales regarding fear/avoidance, malevolence, and authoritarianism (Table 2).

## 3. Results

### 3.1. Descriptive Statistics

Table 3 shows the normative values obtained in the general sample for all the analyzed variables. The subscales demonstrated good reliability and validity, and in this case, also a normal distribution, with skewness and kurtosis values ranging between +1 and −1. The malevolence scale is an exception as it shows distribution values slightly above the norm.

### 3.2. Confirmatory Factor Analysis

Here is the path diagram for the CFA of the PPMI scale (Figure 1), showing how the four areas (fear/avoidance, malevolence, authoritarianism, unpredictability) load onto a single latent factor (PPMI). The diagram includes the loading values for each observed variable, providing a clear visual representation of the factor model. The points represent the variables, and the arrows indicate the direction and magnitude of each variable’s loading on the factors.

The structural equation model (SEM) diagram depicts relationships among four latent constructs labeled “FEA” (likely representing fear/avoidance), “MAL” (malevolence), “AUT” (authoritarianism), and “UNP” (unpredictability). Each construct is represented by a circular node, with double-headed arrows between them indicating correlations. The correlations are strong, with values such as 0.67 between FEA and MAL, 0.80 between MAL and AUT, 0.64 between FEA and AUT, and 0.72 between AUT and UNP, reflecting interconnectedness among these prejudices.

The constructs are linked to observed variables (items labeled P1, P2, etc.), with single-headed arrows showing factor loadings. These loadings indicate the degree to which each item is associated with its latent construct, with values around 0.6–0.8, showing moderate to strong relationships. Some loadings are lower, such as 0.50 or 0.40, which suggests weaker associations for certain items. Additionally, error terms for each observed variable (seen as circular arrows looping back) provide a sense of measurement error, capturing the variability in responses not explained by the latent construct. This diagram overall reflects a confirmatory factor analysis structure, validating how well each item aligns with its designated latent construct and illustrating the inter-relationships among these psychological factors.

As can be seen in Table 4, the baseline chi-square (7093.36, df = 351) serves as a reference for incremental fit indices like CFI and TLI. The Akaike information criterion (AIC) and Bayesian information criterion (BIC) values (47,433.28 and 47,771.15, respectively) evaluate model parsimony, with lower values indicating a better fit, but their interpretation is relative to alternative models. The log-likelihood values (−23,634.64 for the model and −23,243.37 for the unrestricted model) are useful for comparing nested models, highlighting the relative fit of the tested model. Lastly, the RMSEA “not-close fit” tests evaluate whether the model fit is not close, with significant *p*-values (e.g., 3.34 × 10^−11^ for the H_0_ = 0.08) supporting a good fit. These measures complement the primary indices and reinforce the overall reliability of the model.

### 3.3. Discriminant Validity

The correlations between the PPMI subscales (fear/avoidance, malevolence, authoritarianism, and unpredictability) and BIDR self-deceptive enhancement (SDE) and impression management (IM) were generally low to moderate. Specifically, PPMI subscales showed significant but moderate correlations with SDE (*r* ranging from 0.258 to 0.307, *p* < 0.001), suggesting a moderate association but not strong enough to indicate an overlap in constructs. In contrast, the correlations with IM were weaker (*r* ranging from −0.002 to 0.098), with only one significant association for Authoritarianism (*r* = 0.098, *p* = 0.037). These weak correlations with IM suggest minimal influence of socially desirable responding on the PPMI subscales, further supporting discriminant validity.

To complement these findings, the heterotrait/monotrait ratio (HTMT) was calculated, with the following formula:$$HTMT=\frac{Average\;correlations\;within\;items\;of\;the\;same\;construct-Monotrait}{Average\;correlations\;between\;items\;of\;different\;constructs-Heterotrait}$$

The average monotrait correlations (within PPMI subscales) were 0.285, while the average heterotrait correlations (between PPMI subscales and BIDR components) were 0.062. The resulting HTMT ratio was 0.22, which is well below the recommended threshold of 0.85, indicating strong discriminant validity.

## 4. Discussion

The present study on the validity of the PPMI-IT scale is consistent with and expands upon the existing literature, particularly the work by [14] ([14]). Kenny et al.’s original study identified four dimensions of prejudice—fear/avoidance, malevolence, authoritarianism, and unpredictability—underpinning biases toward individuals with mental illnesses. This study validated the translated Italian version (PPMI-IT) and confirmed its four-factor structure using both exploratory and confirmatory factor analyses, which align with Kenny et al.’s findings.

The reliability analysis revealed higher Cronbach’s alpha values for the PPMI-IT subscales (e.g., fear/avoidance α = 0.91 vs. 0.89 in the original; malevolence α = 0.80 vs. 0.73), indicating improved internal consistency. These results suggest that the Italian adaptation retains and possibly enhances the scale’s psychometric properties, particularly for constructs such as malevolence and authoritarianism, which were weaker in the original version.

The confirmatory factor analysis showed strong fit indices (e.g., RMSEA = 0.060, CFI = 0.928, TLI = 0.914, SRMR = 0.060), consistent with Kenny et al.’s validation. The intercorrelations between the subscales (e.g., fear/avoidance and authoritarianism, r = 0.64) were also comparable to those reported by Kenny et al., further supporting the coherence of the theoretical structure. These results validate the scale’s construct validity while confirming its applicability in a new cultural context.

Discriminant validity findings also align with theoretical expectations. The moderate correlations between PPMI subscales and self-deceptive enhancement (SDE) (ranging from r = 0.258 to r = 0.307) and the weaker, mostly non-significant, correlations with impression management (IM) (r = −0.002 to r = 0.098) suggest the scale effectively measures prejudice constructs distinct from socially desirable responding. The calculated HTMT ratio of 0.22 is well below the threshold of 0.85, further confirming discriminant validity. This aligns with the original validation, reinforcing the scale’s ability to measure distinct constructs.

Understanding social desirability is crucial for interpreting findings. For example, a low reported incidence of prejudice in a study may not reflect societal progress but rather an increased awareness of social norms discouraging prejudiced responses.

This result is surprising since traditional self-report measures are limited in their ability to capture unconscious biases. For this reason, tools like the Implicit Association Test (IAT) and physiological measures (e.g., galvanic skin response, fMRI) are increasingly used and preferred, but these methods are resource-intensive and not immune to criticism regarding validity.

In conclusion, the results of this study are consistent with the existing literature, demonstrating that the PPMI-IT scale is a valid and reliable instrument for measuring prejudice toward individuals with mental illnesses in the Italian context. The findings contribute to the growing body of research on stigma and provide a robust tool for cross-cultural comparisons and interventions targeting mental health stigma.

### 4.1. Cultural and Societal Implications

Stigma toward mental illness in Italy is deeply rooted in a combination of historical, cultural, and structural factors. The legacy of institutionalization, particularly prior to the Basaglia Law of 1978, contributed to the perception of individuals with mental disorders as dangerous or socially deviant ([16]). Although the law marked a significant shift toward community-based care, the cultural residue of asylum-based psychiatry continues to influence public attitudes ([6]).

Additionally, widespread cultural beliefs often associate mental illness with personal weakness or moral failure, further perpetuated by stereotypical media portrayals and limited mental health education ([7]; [22]).

Structural stigma remains a significant barrier, with underfunded services, limited access to care, and discrimination in employment ([27]) and housing reinforcing social exclusion ([26]). Culturally, psychological distress is often experienced as a source of shame and silence, shaped by traditional values that emphasize family honor, emotional restraint, and personal strength ([28]). Within this context, religious beliefs can play an ambivalent role: on the one hand, the solidarity and compassion promoted by Catholic doctrine may foster inclusion; on the other hand, more conservative interpretations may associate mental suffering with a lack of faith, personal guilt, or spiritual trials, thereby contributing to stigmatization ([20]).

Finally, as demonstrated by recent literature, individuals with mental illness frequently internalize societal prejudices, which leads to self-stigma, reduced self-esteem, and reluctance to seek help, thereby perpetuating a cycle of marginalization ([15]).

### 4.2. Limitations

While the present study provides strong evidence for the validity and reliability of the PPMI scale, several limitations should be noted.

First, the study relies on self-reported data, which may introduce biases such as social desirability or self-deceptive enhancement, despite efforts to assess these influences through the BIDR scales. Although the correlations and HTMT ratio indicated discriminant validity, the association with self-deceptive enhancement suggests the potential role of social desirability that deserves further investigation.

Second, the sample, although sufficiently wide, may not be fully representative of the general population as it was not stratified by demographic or cultural factors that could influence biases toward individuals with mental health issues. Future research should examine the PPMI scale across diverse cultural and socioeconomic groups to establish its broader applicability and cross-cultural validity.

Lastly, while the model fit indices demonstrate an excellent fit, the reliance on confirmatory factor analysis (CFA) assumes that the hypothesized structure is correct without exploring alternative models. Future studies could use alternative methods, such as exploratory structural equation modeling (ESEM), to confirm whether the four-factor structure is optimal or if refinements are necessary.

Despite these limitations, the study significantly contributes to validating a scale capable of measuring biases toward individuals with mental health issues.

## 5. Conclusions

The analysis conducted confirms the validity of the Italian PPMI and its usefulness in measuring prejudices toward individuals with mental health issues. The four-factor structure is robustly supported by the data, indicating that the scale effectively and distinctly covers various aspects of mental health prejudice. The fit indices demonstrate excellent adherence to theoretical expectations, suggesting that the scale performs well within the Italian context.

Moreover, the analysis of correlations and the heterotrait/monotrait (HTMT) ratio provides strong evidence confirming the discriminant validity of the PPMI scale against social desirability. This demonstrates that the measurements taken with the PPMI scale are not significantly influenced by individuals’ tendencies to respond in a socially desirable manner, thus making the data more reliable and truthful.

Therefore, it can be stated that the PPMI scale is a valid and reliable tool for research and clinical application in the field of mental health. This tool allows researchers and clinicians to identify and quantify levels of prejudice towards mental health in a specific and detailed manner, facilitating targeted interventions and informing public policies to effectively counter the stigma associated with mental illness.

The validation of the PPMI scale thus provides a valuable resource for better understanding how prejudices impact the treatment and perception of individuals with mental disorders in Italian society. The use of this tool can significantly improve intervention strategies, increasing cultural sensitivity and competence in healthcare and social services, thereby reducing barriers to treatment and improving outcomes for affected individuals.

## Figures and Tables

**Figure 1 ejihpe-15-00126-f001:**
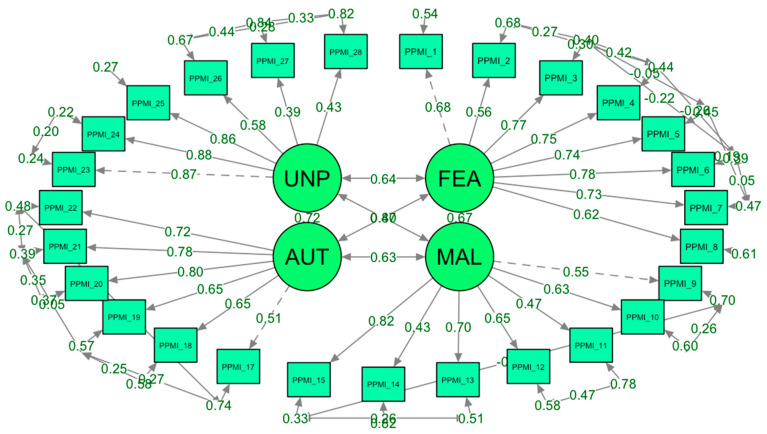
The path diagram for the CFA of the PPMI scale.

**Table 1 ejihpe-15-00126-t001:** PPMI structure and scoring procedure.

Subscale	Description
*Fear/Avoidance*	The sum of the scores for items 1, 2, 3, 4, 5 (R), 6 (R), 7 (R), and 8 (R).
*Malevolence*	The sum of the scores for items 9, 10, 11, 12, 13 (R), 14 (R), 15 (R), and 16 (R).
*Authoritarianism*	The sum of the scores for items 17, 18, 19, 20 (R), 21 (R), and 22 (R).
*Unpredictability*	The sum of the scores for items 23, 24, 25, 26 (R), 27 (R), and 28 (R).

*Note:* (R) indicates reverse-coded items.

**Table 2 ejihpe-15-00126-t002:** Comparison of Cronbach’s alpha values between the original version and the Italian version.

PPMI Scale/Subscale	Cronbach’s Alpha PPMI Original Version ([14])	Cronbach’s Alpha PPMI-IT (Present Study)
Fear/Avoidance	0.89	0.91
Malevolence	0.73	0.80
Authoritarianism	0.72	0.79
Unpredictability	0.86	0.82

**Table 3 ejihpe-15-00126-t003:** Descriptive statistics.

PPMI-IT Subscale	Minimum	Maximum	Mean	St. Dev.	Skewness	Kurtosis
Fear/Avoidance	8	70	32.09	14.216	0.367	−0.487
Malevolence	8	63	16.24	9.656	1.706	3.251
Authoritarianism	6	54	26.51	11.486	0.309	−0.455
Unpredictability	6	54	32.86	9.911	0.143	0.205

**Table 4 ejihpe-15-00126-t004:** Fit indices.

Fit Index	Value
Chi-square (χ^2^)	782.54
Degrees of freedom (df)	296
*p*-value	<0.001
CFI (Comparative Fit Index)	0.928
TLI (Tucker–Lewis Index)	0.914
RMSEA (Root Mean Square Error of Approximation)	0.060
RMSEA 90% CI	[0.055, 0.065]
*p*-value for RMSEA close fit	0.0007
SRMR (standardized root-mean-square residual)	0.060

## Data Availability

Access to the data is available upon reasonable request. For further inquiries, please contact amrizzo@unime.it.

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
