# Peer review of "The Prejudice Towards People with Mental Illness Scale: Psychometric Properties of the Italian Version (PPMI-IT)"

_ejihpe, 2025, doi:10.3390/ejihpe15070126_

Round 1
Reviewer 1 Report
Comments and Suggestions for Authors
Dear Authors,
Thank you for providing me with the opportunity to read this interesting paper. Below, I have listed my comments:
1) The PPMI scale is introduced well, but more could be done to explain why this scale is particularly suited to filling the gaps in existing tools. Right now, it reads as a default solution, rather than a theoretically superior choice. What makes the PPMI scale theoretically robust? How its dimensions map onto the conceptual definition of prejudice you’re using.
2) There is a missed opportunity to comment on how cultural norms in Italy might affect expressions of prejudice differently from other contexts (e.g., family-centered care, Catholic values, mental illness taboos)
For the rest of the sections, I have no comments to make. You’ve written a solid, professional Discussion and Conclusion section.
I hope this feedback is helpful.
Author Response
Response to Reviewer Comments
We thank the reviewer for their thoughtful and constructive feedback, which has helped us improve the clarity and depth of our manuscript. Please find our responses below:
Justification for the Use of the PPMI Scale
We appreciate the suggestion to better articulate the theoretical robustness of the PPMI scale. In the revised manuscript, we have expanded the rationale for its use by highlighting how it addresses limitations in previous tools by capturing both affective and cognitive components of prejudice, making it particularly suited for large population studies.
Cultural and Societal Implications in the Italian Context
We agree that cultural norms play a significant role in shaping expressions of prejudice. In response, we have added new literature and discussion on how specific Italian cultural factors—such as the influence of Catholic values, law, employment, social media, and self-stigma—may influence attitudes toward individuals with mental health conditions. This addition strengthens the contextual relevance of our findings and supports the need for culturally sensitive stigma-reduction strategies.
We are grateful for the reviewer’s positive comments on the Discussion and Conclusion sections and for the helpful suggestions that have enhanced the manuscript.
Reviewer 2 Report
Comments and Suggestions for Authors
This is a very well-written paper, with competently analysed findings and clear presentation. I have no any major comments to raise, although there are a few minor points for consideration.
1. I believe the authors' use of exploratory factor analysis (more precisely, principal components analysis) could be better justified. This technique is generally used when the expected factor structure is unknown. Given that the authors have hypothesised a specific four-factor structure, they should, in my view, instead use only confirmatory factor analysis, as their primary goal is to determine whether the Prejudice Towards People With Mental Illness Scale functions as expected (i.e., has this hypothesised four-factor structure) within the Italian context.
2. The authors should check carefully their writing as there is some repetition of values on page 9, such as: "e.g., Fear/Avoidance and Authoritarianism, r=0.64r = 0.64r=0.64)"
3. The authors should avoid one-paragraph sentences in the paper.
Author Response
Response to Reviewer Comments
We sincerely thank the reviewer for their thoughtful and constructive feedback. We have carefully considered all suggestions and made the following revisions:
Exploratory Factor Analysis (EFA): We agree with the reviewer’s observation regarding the use of EFA. In the revised manuscript, we have removed the exploratory factor analysis and focused solely on confirmatory factor analysis (CFA), as our primary goal is to test the hypothesised four-factor structure of the Prejudice Towards People With Mental Illness Scale within the Italian context.
Repetition of Values: We have carefully reviewed the manuscript and corrected the typographical error on page 9, where the correlation value was unnecessarily repeated.
One-Paragraph Sentences: We have revised the manuscript to avoid overly long one-paragraph sentences, improving readability and clarity throughout the text.
We appreciate the reviewer’s positive assessment of our work and their helpful suggestions, which have contributed to strengthening the quality of the paper.